# System, Method and Software for Calculation of a Cannabis Drug Efficiency Index for the Reduction of Inflammation

**DOI:** 10.3390/ijms22010388

**Published:** 2020-12-31

**Authors:** Nicolas Borisov, Yaroslav Ilnytskyy, Boseon Byeon, Olga Kovalchuk, Igor Kovalchuk

**Affiliations:** 1Moscow Institute of Physics and Technology, 9 Institutsky lane, Dolgoprudny, Moscow Region 141701, Russia; nicolasborissoff@gmail.com; 2Department of Biological Sciences, University of Lethbridge, Lethbridge, AB T1K 3M4, Canada; slava.ilyntskyy@uleth.ca (Y.I.); bbyeon@gmail.com (B.B.); olga.kovalchuk@uleth.ca (O.K.); 3Pathway Rx., 16 Sandstone Rd. S., Lethbridge, AB T1K 7X8, Canada; 4Biomedical and Health Informatics, Computer Science Department, State University of New York, 2 S Clinton St, Syracuse, NY 13202, USA

**Keywords:** cannabis drug efficiency index, signaling pathway impact analysis, anti-inflammatory properties

## Abstract

There are many varieties of *Cannabis sativa* that differ from each other by composition of cannabinoids, terpenes and other molecules. The medicinal properties of these cultivars are often very different, with some being more efficient than others. This report describes the development of a method and software for the analysis of the efficiency of various cannabis extracts to detect the anti-inflammatory properties of the various cannabis extracts. The method uses high-throughput gene expression profiling data but can potentially use other omics data as well. According to the signaling pathway topology, the gene expression profiles are convoluted into the signaling pathway activities using a signaling pathway impact analysis (SPIA) method. The method was tested by inducing inflammation in human 3D epithelial tissues, including intestine, oral and skin, and then exposing these tissues to various extracts and then performing transcriptome analysis. The analysis showed a different efficiency of the various extracts in restoring the transcriptome changes to the pre-inflammation state, thus allowing to calculate a different cannabis drug efficiency index (CDEI).

## 1. Introduction

In the twentieth century, enormous strides have been made in combatting various diseases. Treatment of chronic diseases still remains to be more challenging than acute ones. This is often due to substantial differences in individual responses to known drugs. Over the last few decades, with the advent of genomics and epigenomics, research has focused on the development of personalized medicine. A need has arisen to develop diagnostic tools for use in the characterization of personalized aspects of chronic diseases.

Intracellular signaling pathways (SPs) regulate numerous processes involved in normal and pathological conditions, including development, growth, aging and cancer. Many intracellular signaling pathways or maps are available at online websites. The information relating to signaling pathway activation (SPA) can be obtained from the massive proteomic or transcriptomic data [1]. Although the proteomics analysis may be somewhat closer to the biological function of SPA, the transcriptomics analysis is far more feasible in terms of performing experimental tests and analyzing the data.

The transcriptomic methods like next-generation sequencing (NGS) or microarray analysis of RNA can routinely determine the expression levels for all or virtually all human genes [2]. Transcriptome profiling may be performed from a minute amount of the tissue sample, which does not necessarily need to be fresh, as it can also be done for the clinical formalin-fixed, paraffin-embedded (FFPE) tissue blocks. For the molecular analysis of cancer, gene expression can be interpreted in terms of abnormal SPA features of various pro- and antimitotic signaling pathways [3]. Such an analysis may improve any decision-making process for treatment strategy selection by the clinician.

Pro- and antimitotic SPs that determine various stages of cell cycle progression remained in the spotlight of computational biologists for more than a decade [4,5]. Today, hundreds of SPs and related gene product interaction maps that show sophisticated relationships between the individual molecules are catalogued in various databases, such as UniProt [6], HPRD [7], QIAGEN SABiosciences, WikiPathways [8], Ariadne Pathway Studio [9], SPIKE [10], Reactome [11], KEGG [12], etc.

Many bioinformatics tools have been developed that analyze SPs. One group of bioinformatic approaches integrated the analysis of transcriptome-wide data with the models employing the mass action law and Michaelis–Menten kinetics [13]. However, these methods, which have developed during the last fifteen years, remained purely fundamental until recently, primarily because of the multiplicity of interaction domains in the signal transducer proteins that enormously increase the interactome complexity [14]. Secondly, a considerable number of unknown free parameters, such as kinetics constants and/or concentrations of protein molecules, significantly complicated the SPA analysis. Yizhak et al. (2013) suggested that the clinical efficiency of several drugs, e.g., geroprotectors, may be evaluated as the ability to induce the kinetic models of the pathways into the steady state [13]. However, protein–protein interactions were quantitatively characterized in detail only for a tiny fraction of SPs. This approach is also time-consuming, since to process each transcriptomic dataset, it requires extensive calculations for the kinetic models [13].

In addition, all the contemporary bioinformatical methods that were proposed for digesting large-scale gene expression data, followed by recognition and analysis of SPs, have an important disadvantage: they do not allow tracing the overall pathway activation signatures and quantitively estimate the extent of the SPA [13,15]. This may be due to a lack of the definition of the specific roles of the individual gene products in the overall signal transduction process, incorporated in the calculation matrix used to estimate the SPA.

Thus, there remains an unmet, urgent and increasing need to provide effective personalized non-toxic disease therapies, as well as models for selecting a personalized optimal therapy for an individual.

Here, we propose a method for quick, informative and large-scale screening of changes in SPA in cells and tissues. These changes may reflect various conditions, such as differences in physiological state, aging, disease, treatment with drugs, media composition, additives, etc. One of the potential applications of SPA studies may be to utilize mathematical algorithms to identify and rank the medicines based on their predicted efficacy. In this paper, we give examples of the analysis of transcriptomics data for three sets of experiments involving the analysis of the anti-inflammatory properties of cannabis extracts. Such a method can potentially be used for the detection of the efficiency of various other botanical extracts or single compounds, although the versatility of the method will have to be confirmed using other data sets.

## 2. Methods

### 2.1. Cannabis Drug Efficiency Index (CDEI)

#### 2.1.1. Input and Output Data for the CDEI Metric

Several methods were proposed for the assessment of drug efficiency based on gene/protein expression [16,17,18,19] or mutation patterns [20,21,22]. Unfortunately, most such methods are either proprietary or employ machine learning on preceding cases [23,24,25,26]. So, for evaluating a cannabis drug’s individual action, we have suggested a novel approach, the cannabis drug efficiency index (CDEI).

CDEI is our original approach for assessing the efficiency of a cannabis drug’s application regarding individual persons. CDEI is calculated based on high-throughput gene expression profiling and a signaling pathway topology, by comparison of gene expression and signaling pathway profiles between case and normal reference states for prediction of drug action in individual cases.

As input data, the CDEI operates with the results of various “omics” data stemming from the cells of individual patients and the healthy individuals. These data may include transcriptomic (e.g., performed with either next-generation sequencing or microarray mRNA hybridization), non-coding-RNAomic, proteomic and epigenomic data, etc.

The data of the full mRNA/protein abundance are integrated by the CDEI into the assessment values for activation of different cellular pathways (signalome).

The topology for the signaling, metabolic and cytoskeleton pathways, etc., were obtained from the database of QIAGEN SABiosciences (URL: https://www.qiagen.com/ru/shop/genes-and-pathways/pathway-central/).

#### 2.1.2. SPIA (Signaling Pathway Impact Analysis): A Method for Assessment of Signaling Pathway Activation

The high-throughput gene expression data were converted according to our CDEI approach to more aggregated values of the molecular pathway activities, using the signaling pathway impact analysis (SPIA) method [27]. Among the other methods for gene-expression-based assessment of signaling pathway activation, including TAPPA [28], topology-based score (TB) [29], Pathway-Express (PE) [1], and OncoFinder [1], the SPIA [27] approach showed the best statistical performance during the comparison between pathway-based and gene-based values (Figure 1; taken from [30]).

Imagine a pathway graph, *G*(*V, E*) where $V=\;\left\{g_{1},\;g_{2},\;\ldots ,\;g_{n}\right\}$ is the set of graph nodes (vertices), and $E=\left\{\left(g_{i},g_{j}\right)\;|\;\mathrm{genes}\;g_{i}\;\mathrm{and}\;g_{j}\;\mathrm{interact}\right\}$ is the set of graph edges. The adjacency matrix is defined as $a_{ij}=1$, if i=j or $\left(g_{i},g_{j}\right)\in E$, and $a_{ij}=0$, if $\left(g_{i},g_{j}\right)\notin E$.

Consider then the values, called perturbation factors (*PF*), for the all genes *g* of the pathway *K*,
$$PF\left(g\right)=\;\Delta E\left(g\right)+\underset{\gamma \in U_{g}}{\sum }\beta_{\gamma g}\cdot \frac{PF\left(\gamma \right)}{n_{down}\left(\gamma \right)}.$$
Here, ΔE(g) is the signed log-fold-change of the gene *g* expression level in a given sample compared with the average value for the pool of normal samples. The latter term expresses the summation over all the genes γ that belong to the set *U_g_* of the upstream genes for the gene *g*. The value of *n_down_* (γ) denotes the number of downstream genes for gene γ. The weight factor $\beta_{yg}$ indicates the interaction type between γ and *g*: $\beta_{yg}=1$ if γ activates *g*, and $\beta_{yg}=1$ when γ inhibits *g*. The search for upstream/downstream genes is performed according to the depth-first search method [31].

To obtain an estimator for pathway perturbation that is positive for an upregulated and negative for a downregulated pathway, use the second term in the formula for the perturbation factor (PF) from the previous paragraph, resulting in the accuracy value, Acc(g)=PF(g)−ΔE(g). It can be shown that this accuracy vector may be expressed as follows: $Acc=B\cdot {\left(I-B\right)}^{-1}\cdot \Delta E$, where
$$B=\left(\begin{array}{ccc} \frac{\beta_{11}}{n_{down}\left(g_{1}\right)} & \cdots & \frac{\beta_{1n}}{n_{down}\left(g_{n}\right)}\\ \vdots & \ddots & \vdots \\ \frac{\beta_{n1}}{n_{down}\left(g_{1}\right)} & \cdots & \frac{\beta_{nn}}{n_{down}\left(g_{n}\right)} \end{array}\right),$$

**I** is the identity matrix, and
$$\Delta E=\left(\begin{array}{c} \begin{array}{c} \Delta E\left(g_{1}\right)\\ \ldots \end{array}\\ \Delta E\left(g_{n}\right) \end{array}\right).$$
The overall score for the pathway perturbation was calculated as $SPIA=\;\underset{g}{\sum }Acc\left(g\right)$ [27].

#### 2.1.3. Calculation of the Cannabis Drug Efficiency Index (CDEI)

The SPIA-based CDEI metric was then calculated according to the following algorithm:Obtain the signaling pathway impact analysis (SPIA) for each drug for each biological pathway.Calculate the values of the pathway weight (*w_p_*) factor as follows: For pathways with a positive mean SPIA score of the case samples, *w_p_* = ((number of case samples with positive SPIA score)/(total number of case samples)). For pathways with a negative mean SPIA score of the case samples, *w_p_* = ((number of case samples with negative SPIA score)/(total number of case samples)).Adjust the mean SPIA score of each pathway by the weight factor, SPIA_μ_ = mean(SPIA)·*w_p_*.Perform Student’s t-test if the values of SPIA_μ_ for the pool of case samples are different from 0 (for the pool of control samples, the values of SPIA_μ_ are clearly equal to 0). During the Student’s t-test, the following case classes are taken into account: the untreated case (*U*), i.e., the pathological state before drug application, should be far from the control (*C*).-Treated case (*T*), i.e., the pathological state after drug application, should be close to the control;-The following values are the results for such calculations: |*t_U_*| = absolute t-value for the Student’s t-test for *U*-vs.-*C* profiles;-|*t_T_*| = absolute t-value for the Student’s test for *T*-vs.-*C* profiles.Calculate the cannabis drug efficiency index (CDEI) for each drug for a specific disease, wherein CDEI = 2 ((|*t*_U_|/(|*t*_T_| + |*t*_U_|) − 0.5).Rank the drugs according to highest CDEI for a group of individual patients.

Note that our value of CDEI has the following properties:-CDEI is a value between −1 and 1;-CDEI is 0 if |*t_T_*| and |*t_U_*| are the same, which means no drug efficiency;-CDEI is 1 if |*t_T_*| is 0, which means the perfect efficiency;-CDEI is a value greater than 0 if |*t_T_*| is smaller than |*t_U_*|, which means a positive efficiency;-CDEI is a value less than 0 if |*t_T_*| is larger than |*t_U_*|, which means a negative efficiency.

Note also that the mean score of the case samples in each pathway is first calculated/adjusted and then the set of the mean scores in each data set are t-tested for their difference from 0 (i.e., a one-sample *t*-test). So, a t-statistic is calculated for each dataset and the CDEI metric is one value for each case sample dataset.

The CDEI calculations based on the SPIA values were done using R script.

### 2.2. Experiments Used for Validation of the CDEI

The CDEI metric was tested on several datasets, as explained below in several examples.

#### 2.2.1. Plant Growth

All cannabis plant hybrids were generated and grown in a licensed facility at the University of Lethbridge. Various cultivars, numbered #1, #4, #7, #8, #9, #12, #13, #45, #114, #129, #130, #157, #167, #169 and #274, were used for the analysis. All hybrids represent individual lineages with different levels of cannabinoids (data not shown). Cuttings from the mother plants of the same age (~6 months) were made and allowed to root. After rooting (~10 days), plantlets were acclimated for another week and then were transplanted to a larger pot. Four plants per variety were grown at 22 °C for 18 h in light and 6 h in the dark, for 4 weeks, and then transferred to the chambers for a 12 h light/12 h dark regime to promote flowering. Plants were grown to maturity and flowers were harvested and dried. Approximately 5 g of the dry, lower samples from each of four plants per variety were combined and then used for extraction.

#### 2.2.2. Crude Extract Preparation Using Solvent

Three grams of plant tissue was ground to a powder using a fine coffee grinder and the powdered plant tissue were weighed using an analytical balance. Plant material was placed inside a 250 mL Erlenmeyer flask. A total of 100 mL of ethyl acetate was poured into the flask containing the plant material. The flasks were then wrapped with tin foil and shaken continuously (120 rpm) in an incubator at 21 °C overnight and in the dark.

After overnight solvent extraction, the extracts were filtered through cotton into a 100 mL round bottom flask. The extracts were concentrated to around 2–3 mL using a rotary vacuum evaporator. The extracts were then transferred to a tared 3-dram vial (cat# 60975L Kimble obtained from Fisher Scientific). The leftover solvent was evaporated to dryness in an oven overnight at 50 °C to eliminate the solvent completely. The mass of each crude extract was recorded. All extractions were repeated twice.

#### 2.2.3. Analysis of Cannabinoid Content

The levels of cannabinoids were analyzed using an Agilent Technologies 1200 Series HPLC system. The extract stocks were prepared from the crude extracts whereby 3–6 mg of crude extract was dissolved in DMSO (dimethyl sulfoxide anhydrous, Life Technologies) to reach a 60 mg/mL final concentration and stored at −20 °C. The appropriate cell culture media (RPMI + 10% FBS or EMEM + 10% FBS) were used to dilute the 60 mg/mL stock to make a working medium containing 0.01 mg/mL. The extracts were sterilized using a 0.22 µm filter. The composition of each extract is shown in Table 1.

#### 2.2.4. Preparation of the Cannabis Extract for Experimental Analysis Using Human Cells and Tissues

The stocks were prepared weighing 3–6 mg of crude extract into a micro centrifuge tube. The crude extract was dissolved in DMSO (dimethyl sulfoxide anhydrous from Life technologies cat # D12345) to reach a 60 mg/mL final concentration and stored at −20 °C. For the assay, a different amount of stock material (from 1 to 20 µL) was added to 21 mL of medium to make a working extract. Different concentrations were tested for cell/tissue toxicity and one concentration was chosen for further work (data not shown). Specifically, to obtain “low” (0.007 mg/mL) and “high” (0.015 mg/mL) functional concentrations, 2.45 µL or 5.25 µL of stock extract (60 mg/mL) were added to 21 mL of medium, respectively. The final concentration of DMSO was 0.012% and 0.025% in the “low” and “high” extract concentrations, respectively. The extracts were sterilized using a 0.22 µm filter.

#### 2.2.5. Example #1

Human EpiDermFT 3D skin tissues (MatTek Life Sciences, Ashland, MA, USA) were exposed to 7000 ergs UVC to induce inflammation and then, 24 h after exposure, treated with extracts of several cannabis cultivars (#4, #8, #12 and #13) via their addition to the tissue growth media and incubated for another 24 h. Untreated sample (*U*) had DMSO added to the media instead of extracts. The control (*C*) sample had not been exposed to UVC. All samples were collected 24 h after the extracts were added and were used for mRNA extraction. All samples were done in triplicate.

RNA samples were extracted using TRIzolTM (Sigma Aldrich, St. Louis, MO, USA) according to the manufacturer’s instructions. The cDNA fragment libraries were prepared using the TruSeq Stranded mRNA library preparation kit (Illumina, San Diego, CA, USA) with polyA selection, as described in the manual. The high-throughput gene expression profiles were obtained using the Illumina mRNA next-generation sequencing platform NextSeq500. All of the library preparations and sequencing runs were completed using the same protocol, by the same technician, and on the same sequencing instrument.

#### 2.2.6. Example #2

Human MatTek’s 3D EpiOral tissues (MatTek Life Sciences, Ashland, MA, USA) were equilibrated for 24 h, then the culture medium was replaced and the tissues incubated for another 24 h. Tissues were then exposed for 24 h to TNFα (40 ng/mL) to promote inflammation or to DMSO only. Tissues were then treated with various cannabis extracts (#1–#9) that were added to the media for 24 h. The control sample was exposed to DMSO only. Samples were then harvested for mRNA extraction and sequenced as in Example #1. All samples were done in triplicate.

#### 2.2.7. Example #3

Human MatTek’s 3D EpiIntestinal tissues (MatTek Life Sciences, MA) were equilibrated for 24 h, then the culture medium was replaced, and the tissues incubated for another 24 h. Tissues were then exposed for 24 h to TNFα (40 ng/mL) or to DMSO only. Tissues were then treated with various cannabis extracts (#1, #2, #3, #4, #5, #6, #9, #10 and #11) that were added to the media for 24 h. The control sample was exposed to DMSO only. Samples were then harvested for mRNA extraction. Sequencing and data analysis were performed as in Example #2. All samples were done in triplicate.

#### 2.2.8. Bioinformatics Workflow

The bioinformatics workflow for the CDEI calculation is shown in Figure 2. Basecalling and demultiplexing were done using the CASAVA v.1.9 pipeline (Illumina, San Diego, CA, USA). The quality of the sequencing reads was assessed using FastQC v0.11.5 (https://www.bioinformatics.babraham.ac.uk/projects/fastqc/). The base qualities were over 30 on the Phred scale across all of the samples and no adapter contamination was noted. The reads were mapped to the human genome (GRCh37, Ensembl) downloaded from the Illumina iGenomes repository (https://support.illumina.com/sequencing/sequencing_software/igenome.html). The mapping was done using HISAT v2.0.5 [32] with the following command “*hisat2-q—rna-strandness R—phred33-p 20—known-splicesite <known_splice_sites> -x <index> -U <fastq> -S <sam>*”. The number of reads mapping to features (genes) was calculated using FeatureCounts v.1.6.1 [33], with the following command “*featureCounts -T 20 -s 2 -a <genes.gtf> -o <counts> <sam>*”.

The gene-level read counts were loaded into the R v.3.6.1 statistical language environment. The raw count data were normalized using statistical methods implemented in the DESeq2 Bioconductor package v1.22.0 [34]. The DESeq normalized read counts were used to calculate the CDEI index using CDEI software. Table 2 shows the list of inflammation-related genes used for the CDEI calculation.

## 3. Results

The CDEI software was tested on several datasets, explained below in several examples. We used the transcriptomic data from three different experiments. In the first experiment, we have used the data from human EpiDermFT 3D skin tissues exposed to UVC to induce inflammation and then treated with extracts of several cannabis cultivars. In the second experiment, human EpiOral tissues were treated with TNFα to induce inflammation and then treated with several different extracts. In the third experiment, human EpiIntestinal tissues were treated with TNFα and then treated with several different extracts.

The total number of reads calculated from the three experiments was in the range of 17,700,978 to 45,054,688 with a median of 24,613,834.5 reads per sample. The minimum mapping rate, calculated as a fraction of reads with at least one match to the genome, was at least 93.80% with a median of 97.65%. The fraction of mapped reads assigned to features (genes) was in the range of 72.93–76.00% with a median of 74.86%.

In the first experiment, we have induced inflammation in human EpiDermFT 3D skin tissues by exposing it to UVC. Analysis of the samples from the first experiment revealed that Extract #8 is the most efficient extract in restoring the transcriptome response after the UVC exposure (Table 3). Extract #4 was less efficient, whereas Extract #13 was not efficient. Extract #12 was actually harmful as it has increased the UVC-induced changes in the transcriptome.

In the second experiment, inflammation in 3D EpiOral tissues was established using exposure to TNFα (40 ng/mL) (Table 4). The effect of the extracts on the reversal of the inflammation processes was evaluated using the CDEI.

Ranking of the CDEI scores revealed that Extract #3 was the most efficient; it has restored the TNF-induced transcriptome nearly completely—with a CDEI score of 0.98. Extracts #5, #9 and #2 were also quite efficient, with CDEI scores of 0.90, 0.88 and 0.87, respectively. Extracts #8 and #4 were not very efficient, with CDEI scores of 0.10 and 0.16, respectively.

Finally, in the 3rd experiment, EpiIntestinal tissues were exposed to TNFα and several extracts were used to reverse the transcriptome changes (Table 5). The CDEI score showed that Extract #5 was the most efficient, followed by Extract #6.

Examples of heatmaps, with the differentially expressed genes for Samples #4, #8, #13 and #15, are shown in Figure 3. 

These experiments revealed that different extracts were efficient for different tissues. For example, a comparison of extracts for the reduction of inflammation in EpiOral and EpiIntestinal tissues showed that Extract #115 was equally and highly effective, while Extract #167 was not (Figure 4). At the same time, Extracts #7, #9 and #169 were effective for EpiOral tissues but not for EpiIntestinal tissues (Figure 4). Our attempts to correlate the level of cannabinoids and the efficiency of the extracts did not show any correlation. This is perhaps not surprising, as other molecules in the extracts likely have a strong modulatory effect.

To analyze whether there is a correlation between the cannabinoid content and the activity of the extracts, we have analyzed the amount of THC, CBD, CBG, CBN, total cannabinoids and CBD to THC ratio (Table 1). The correlation analysis showed a moderate positive correlation between the CBD level and CDEI score (0.49) and a strong positive correlation between the CBN level and CDEI score (0.61). No correlation was found between THC and CDEI, CBG and CDEI or the CBD to THC ratio and CDEI.

## 4. Discussion

We developed a methodology to compute a cannabis drug efficiency index (CDEI) across the 241 pathways that contain the genes responding to the cannabis drug. The advantages of our methodology can be summarized as follows.

CDEI explicitly calculates the drug efficiency of cannabis on diseases, using the expression values of the genes. As far as we understand, this is a novel methodology and there is no other methodology or software similar to CDEI to do it.

CDEI evaluates the enrichment scores of the individual pathways in the control, untreated and treated cases. Then CDEI statistically integrates and compares the results from all samples. Finally, CDEI measures the overall drug efficiency of cannabis on diseases.

An important characteristic of the CDEI calculation is that CDEI is not affected by the properties of the input data chosen. CDEI purely measures the efficacy of the cannabis drug by statistically comparing the cannabis-treated and untreated cases with the control.

With the emergence of large-scale methods in genomics, the gene-level transcription changes became an obvious target in the search for biomarkers in complex diseases. Yet, in over a decade of research effort, only a handful of gene or protein biomarkers made its way into clinical practice. For example, as of 2017, 26 gene-based predictive and diagnostic biomarkers were used in cancer medicine [35]. Biomarker discovery based on gene expression signatures is challenging due to low reproducibility when presented with different datasets [36,37] and a high occurrence of stochastic associations [38]. As an alternative to a single gene or gene signature approach, methods that aggregate data across pathways or gene network modules are gaining traction, potentially being more effective tools for biomarker discovery [39,40,41]. Several studies demonstrated that the use of pathway-based biomarkers enables robust predictions of drug response [42,43] and accurate classification of disease types [44].

Considering the importance of pathway analysis, multiple methods were developed to address the task. These methods can be roughly classified as non-topology-based and topology-based [45]. Non-topological methods treat pathways as non-structured lists of genes (gene sets); they take a list of differentially expressed (DE) genes and attempt to determine the probability of observing a given number of DE genes within a pathway. More sophisticated non-topological methods rely on the analysis of gene rankings in the whole dataset, avoiding the selection of target gene lists according to arbitrary thresholds. Topology-based methods take into account the biological reality of pathways by incorporating the data on the type and direction of protein interactions. Not surprisingly, topology-based methods were shown to outperform their counterparts in benchmarking tests [45].

Recognizing the importance of pathway topology, CannSelect uses the SPIA algorithm to calculate pathway activation scores based on gene expression profiles. Previously, SPIA was shown to outperform other pathway analysis methods in terms of data aggregation [30] and in the ability to detect pathways, inducing a specific phenotype [45]. SPIA was also ranked third out of 10 methods used to generate pathway activation scores for machine learning classification in an iPanda benchmarking test [17].

Estimation of pathway-level activities opens a possibility of selecting drugs or other bioactive compounds that could alter the physiological state of the system in the desired direction by changing the activities of perturbed pathways.

This idea is at the core of several computational methods, including Oncofinder [1], GeroScope [46] and iPanda [17]; this is also true for the method presented in this study. Oncofinder is based on the unique computational algorithm that calculates pathway activation strength based on the activating or repressive actions of proteins in the pathway. Pathway activation strength reflects the activation or inhibition of the pathway in a pathological state compared to the healthy controls. It also incorporates the data on drug–target interactions to predict the action of the drugs on specific pathways and rank the drugs based on a predicted efficiency score. The output of Oncofinder is a list of drugs ranked by the predicted efficiency meant to assist in selecting a treatment strategy for a specific patient. GeroScope is an extension of Oncofinder to analyze aging-related pathways and screen chemical compounds that act as geroprotectors. iPanda is a topology-based method that assigns importance coefficient to genes in a pathway derived using a combination of statistical and topological methods [17]. iPanda can be adapted to estimate drug scores following similar principles as in Oncofinder.

In theory, the CDEI approach mechanistically resembles Oncofinder, Geroscope, or iPanda; however, it differs from them in terms of pathway score and drug score algorithms and, more importantly, it is designed with a different purpose in mind. Oncofinder and other similar methods require two conditions—disease and normal state—and serve to facilitate the selection of drugs that could act on pathways altered between these two conditions. The data can be obtained experimentally or downloaded from public repositories. CDEI, on the other hand, requires three conditions—an untreated control (healthy state), pathological state, and pathological state treated with a drug of interest. The output of CDEI is a numerical measure of the ability of a drug to reverse gene expression changes in a pathological state to mimic those in the healthy control. In contrast to Oncofinder, CDEI is useful as a computational step that follows laboratory testing of panels of drugs or other bioactive compounds to select prospective candidates for further investigation.

There are different methodologies to analyze the enrichment of individual pathways. GSEA (Gene Set Enrichment Analysis) finds the enrichment of pathways by focusing on gene sets [47]. SPIA measures the perturbation score on pathways [26]. NEA (Network Enrichment Analysis) outputs the network enrichment scores of the altered gene sets per pathway [48]. However, these methodologies do not further derive an integrated meaningful result (i.e., a drug efficiency index) from the analysis of the individual pathways.

As detailed in the accompanying manual of the software, CDEI provides a convenient graphical user interface software available to users. The CDEI software works on both Windows and Linux systems. It validates the user inputs of the gene expression values and computes the scores of the individual pathways as an intermediate output. Then the software generates the drug efficiency index of cannabis on diseases as the final output.

One of the recent studies using the SPIA method was the report by Franco et al. (2019) [48]. The authors presented a method to identify the biomarkers of drug response and survival in proliferative disease using enrichment over the gene networks. Unlike the CDEI algorithm, it does not calculate drug scores that reflect the overall shift in pathway activity towards a normal (non-diseased) state; also, in their study, the SPIA scores were calculated for individual pathways and correlated with clinical variables of interest (drug response and survival). In the CDEI, the SPIA scores are weighted and summarized following a specific application of a drug screen to the individual sample, which allows a calculation of a drug score and the ranking of drugs by efficiency. In this, respect the CDEI is a tool that allows a selection of a drug for an individual patient. This functionality is not present in Franco’s network approach [48].

Pathway analysis has the potential to improve performance by applying machine learning methods. For example, the representative genes that represent a pathway can be selected by feature selection methods and pathways can be further ranked or weighted based on the representative genes using classification methods [49].

To conclude, the CDEI algorithm represents a multistage process that includes the specific application of a drug screen, integration of pathway databases, weighted summarization of the pathway activation scores and identification of the most efficient drugs/extracts. The bioinformatics part of the CDEI process is wrapped into user-friendly graphical interface software. Currently, the CDEI algorithm was only used for the analysis of the efficacy of cannabis extracts in the reduction of inflammation. It remains to be shown whether it would also be able to rank the cannabis extracts (or extracts of any other medicinal herb or synthetic drug) by their efficacy for other molecular processes, diseases, or conditions. The CDEI algorithm may also be an efficient tool to stratify patients in the clinical studies or clinical trials by their response to a drug, thus providing selection criteria upon transition from Phase 2 to Phase 3 or Phase 4 clinical trials. It will also be very useful in pre-clinical experiments using various cell, tissue or animal models, like recently used analysis of the efficiency of cannabis extracts for the reduction of inflammation [50].

## Figures and Tables

**Figure 1 ijms-22-00388-f001:**
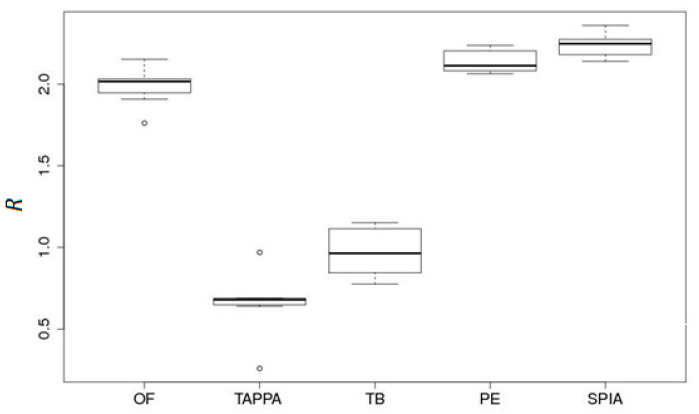
Data aggregation effect *R* for five pathway activation scoring methods (OncoFinder (OF), TAPPA, TBScore (TB), Pathway-Express (PE) and signaling pathway impact analysis (SPIA)) on a renal carcinoma dataset [30].

**Figure 2 ijms-22-00388-f002:**
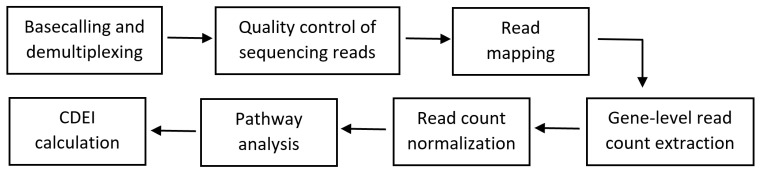
Bioinformatics workflow for the cannabis drug efficiency index (CDEI) calculation.

**Figure 3 ijms-22-00388-f003:**
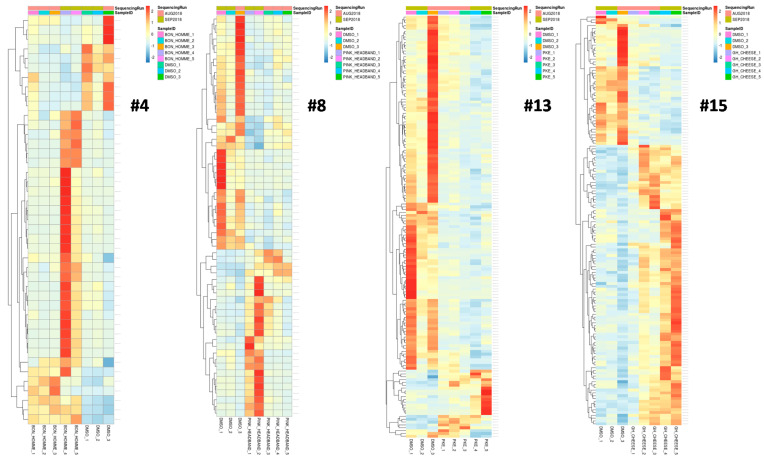
Heatmaps of the differentially expressed genes for Extracts #4, #8, #13 and #15. Five independent samples for the extracts and three independent samples for DMSO (control) are shown. Only genes with a *p*-value < 0.05 are shown.

**Figure 4 ijms-22-00388-f004:**
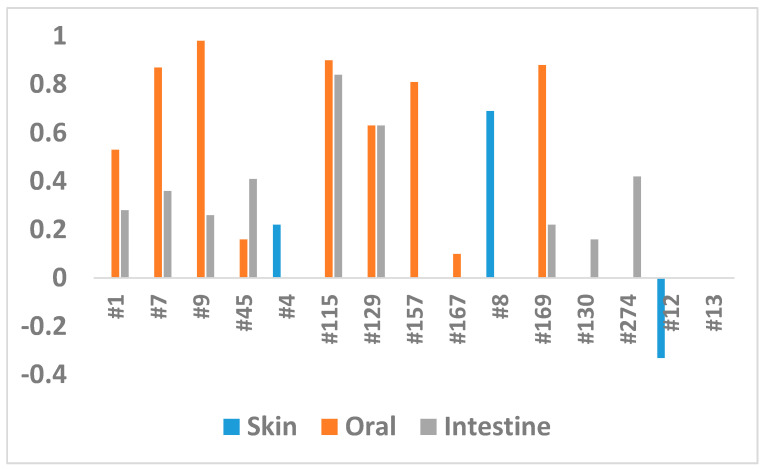
Comparison of the CDEI in three experiments. The Y axis shows the CDEI, while the *X* axis shows the extracts.

**Table 1 ijms-22-00388-t001:** Level of single and total cannabinoids in the flowers and extracts of selected *C. sativa* cultivars.

Flowers, %	THC	CBD	CBGA	CBN	TOTAL Cannabinoids	CBD:THC Ratio
**#1**	0.25	6.79	0.12		7.16	27.16
**#7**	0.21	7.2	0		7.41	34.29
**#9**	0.22	6.91	0.31		7.44	31.41
**#45**	0.03	1.61			1.64	53.67
**#115**	0.3	9.54			9.84	31.80
**#129**	0.28	6.75	0.66		7.69	24.11
**#130**	0.86	2.63	0.31	0.03	3.83	3.06
**#157**	0.2	3.75	0.09	0.15	4.19	18.75
**#167**	0.08	2.25	0.16		2.49	28.13
**#169**	0.2	1.88	0.14		2.22	9.40
**#274**	0.44	9.02	0.31		9.77	20.50
**Extracts, %**	**THC**	**CBD**	**CBGA**	**CBN**	**TOTAL Cannabinoids**	**CBD:THC Ratio**
**#1**	0.88	34.6	0.25	0.12	35.85	39.32
**#7**	1.1	32.9	0.27	0.15	34.27	29.91
**#9**	0.98	32.6	0.97	0.15	34.55	33.27
**#45**	0.44	24.92	0.13	0.14	25.63	56.64
**#115**	1.23	42.52	0.42	0.28	44.45	34.57
**#129**	1.3	35.3	1.2	0.42	38.22	27.15
**#130**	2.43	28.43	0.98	0.18	32.02	11.70
**#157**	0.62	33.5	0.73	0.33	34.85	54.03
**#167**	0.38	24.3	0.29	0.12	24.97	63.95
**#169**	0.67	19.28	0.45	0.18	20.58	28.78
**#274**	0.93	43.81	1.2	0.12	46.06	47.11
**Molarity/µM**	**THC**	**CBD**	**CBGA**	**CBN**	**TOTAL Cannabinoids**	**TOTAL Cannabinoids**
**#1**	0.28	11.00	0.08	0.04		
**#7**	0.35	10.46	0.09	0.05		
**#9**	0.31	10.37	0.31	0.05		
**#45**	0.14	7.92	0.04	0.05		
**#115**	0.39	13.52	0.13	0.09		
**#129**	0.41	11.23	0.38	0.14		
**#130**	0.77	9.04	0.31	0.06		
**#157**	0.20	10.65	0.23	0.11		
**#167**	0.12	7.73	0.09	0.04		
**#169**	0.21	6.13	0.14	0.06		
**#274**	0.30	13.93	0.38	0.04		

Concentration of cannabinoids is shown in percentage of total dry weight (%) or in moles.

**Table 2 ijms-22-00388-t002:** List of genes used for calculation of the level of gene expression in inflammation.

UniProt	Gene Symbol	Gene Name
UniProtKB: Q12979	ABR	Active breakpoint cluster region-related protein
UniProtKB: P13686	ACP5	Tartrate-resistant acid phosphatase type 5
UniProtKB: Q04771	ACVR1	Activin receptor type-1
UniProtKB: P00813	ADA	Adenosine deaminase
UniProtKB: P30542	ADORA1	Adenosine receptor A1
UniProtKB: P29274	ADORA2A	Adenosine receptor A2a
UniProtKB: P29274	ADORA2A	Adenosine receptor A2a
UniProtKB: P29275	ADORA2B	Adenosine receptor A2b
UniProtKB: Q15109	AGER	Advanced glycosylation end product-specific receptor
UniProtKB: Q15109	AGER	Advanced glycosylation end product-specific receptor
UniProtKB: P50052	AGTR2	Type-2 angiotensin II receptor
UniProtKB: P50052	AGTR2	Type-2 angiotensin II receptor
UniProtKB: P23526	AHCY	Adenosylhomocysteinase
UniProtKB: P31749	AKT1	RAC-alpha serine/threonine-protein kinase
UniProtKB: P09917	ALOX5	Arachidonate 5-lipoxygenase
UniProtKB: P02652	APOA2	Apolipoprotein A-II
UniProtKB: Q9NR48	ASH1L	Histone-lysine N-methyltransferase ASH1L
UniProtKB: P00966	ASS1	Argininosuccinate synthase
UniProtKB: Q13315	ATM	Serine-protein kinase ATM
UniProtKB: P30530	AXL	Tyrosine-protein kinase receptor UFO
UniProtKB: P15291	B4GALT1	Beta-1,4-galactosyltransferase 1
UniProtKB: Q9Y5Z0	BACE2	Beta-secretase 2
UniProtKB: Q92560	BAP1	Ubiquitin carboxyl-terminal hydrolase BAP1
UniProtKB: P11274	BCR	Breakpoint cluster region protein
UniProtKB: P46663	BDKRB1	B1 bradykinin receptor
UniProtKB: P22004	BMP6	Bone morphogenetic protein 6
UniProtKB: O00238	BMPR1B	Bone morphogenetic protein receptor type-1B
UniProtKB: Q06187	BTK	Tyrosine-protein kinase BTK
UniProtKB: Q06187	BTK	Tyrosine-protein kinase BTK
UniProtKB: Q06187	BTK	Tyrosine-protein kinase BTK
UniProtKB: Q5T7M4	C1QTNF12	Adipolin
UniProtKB: P01024	C3	Complement C3
UniProtKB: P01024	C3	Complement C3
UniProtKB: Q96GV9	C5orf30	UNC119-binding protein C5orf30
UniProtKB: P13671	C6	Complement component C6
UniProtKB: P01258	CALCA	Calcitonin
UniProtKB: P01258	CALCA	Calcitonin
UniProtKB: Q16602	CALCRL	Calcitonin gene-related peptide type 1 receptor
UniProtKB: P51671	CCL11	Eotaxin
UniProtKB: O00175	CCL24	C-C motif chemokine 24
UniProtKB: P51679	CCR4	C-C chemokine receptor type 4
UniProtKB: P10747	CD28	T-cell-specific surface glycoprotein CD28
UniProtKB: P25942	CD40	Tumor necrosis factor receptor superfamily member 5
UniProtKB: P40200	CD96	T-cell surface protein tactile
UniProtKB: Q9BWU1	CDK19	Cyclin-dependent kinase 19
UniProtKB: Q9UNI1	CELA1	Chymotrypsin-like elastase family member 1
UniProtKB: O15516	CLOCK	Circadian locomoter output cycles protein kaput
UniProtKB: P21554	CNR1	Cannabinoid receptor 1
UniProtKB: P21554	CNR1	Cannabinoid receptor 1
UniProtKB: P34972	CNR2	Cannabinoid receptor 2
UniProtKB: P25025	CXCR2	C-X-C chemokine receptor type 2
UniProtKB: P11511	CYP19A1	Aromatase
UniProtKB: Q9NR63	CYP26B1	Cytochrome P450 26B1
UniProtKB: Q9Y271	CYSLTR1	Cysteinyl leukotriene receptor 1
UniProtKB: Q9NRR4	DROSHA	Ribonuclease 3
UniProtKB: Q1HG43	DUOXA1	Dual oxidase maturation factor 1
UniProtKB: Q1HG44	DUOXA2	Dual oxidase maturation factor 2
UniProtKB: Q9Y6W6	DUSP10	Dual specificity protein phosphatase 10
UniProtKB: Q16610	ECM1	Extracellular matrix protein 1
UniProtKB: P24530	EDNRB	Endothelin receptor type B
UniProtKB: P00533	EGFR	Epidermal growth factor receptor
UniProtKB: P00533	EGFR	Epidermal growth factor receptor
UniProtKB: Q9BQI3	EIF2AK1	Eukaryotic translation initiation factor 2-alpha kinase 1
UniProtKB: P08246	ELANE	Neutrophil elastase
UniProtKB: P29317	EPHA2	Ephrin type-A receptor 2
UniProtKB: P01588	EPO	Erythropoietin
UniProtKB: P03372	ESR1	Estrogen receptor
UniProtKB: P14921	ETS1	Protein C-ets-1
UniProtKB: P15090	FABP4	Fatty acid-binding protein, adipocyte
UniProtKB: O15360	FANCA	Fanconi anemia group A protein
UniProtKB: Q9BXW9	FANCD2	Fanconi anemia group D2 protein
UniProtKB: P30273	FCER1G	High affinity immunoglobulin epsilon receptor subunit gamma
UniProtKB: P30273	FCER1G	High affinity immunoglobulin epsilon receptor subunit gamma
UniProtKB: P30273	FCER1G	High affinity immunoglobulin epsilon receptor subunit gamma
UniProtKB: P30273	FCER1G	High affinity immunoglobulin epsilon receptor subunit gamma
UniProtKB: Q12946	FOXF1	Forkhead box protein F1
UniProtKB: Q9BZS1	FOXP3	Forkhead box protein P3
UniProtKB: P22466	GAL	Galanin peptides
UniProtKB: P23771	GATA3	Trans-acting T-cell-specific transcription factor GATA-3
UniProtKB: Q13304	GPR17	Uracil nucleotide/cysteinyl leukotriene receptor
UniProtKB: P07203	GPX1	Glutathione peroxidase 1
UniProtKB: P36969	GPX4	Phospholipid hydroperoxide glutathione peroxidase
UniProtKB: P81172	HAMP	Hepcidin
UniProtKB: Q30201	HFE	Hereditary hemochromatosis protein
UniProtKB: P14210	HGF	Hepatocyte growth factor
UniProtKB: Q96A08	HIST1H2BA	Histone H2B type 1-A
UniProtKB: P10809	HSPD1	60 kDa heat shock protein, mitochondrial
UniProtKB: P05362	ICAM1	Intercellular adhesion molecule 1
UniProtKB: P14902	IDO1	Indoleamine 2,3-dioxygenase 1
UniProtKB: P14902	IDO1	Indoleamine 2,3-dioxygenase 1
UniProtKB: P22692	IGFBP4	Insulin-like growth factor-binding protein 4
UniProtKB: P22301	IL10	Interleukin-10
UniProtKB: P29460	IL12B	Interleukin-12 subunit beta
UniProtKB: P35225	IL13	Interleukin-13
UniProtKB: P40933	IL15	Interleukin-15
UniProtKB: Q9UHF5	IL17B	Interleukin-17B
UniProtKB: Q96PD4	IL17F	Interleukin-17F
UniProtKB: Q96F46	IL17RA	Interleukin-17 receptor A
UniProtKB: Q9NRM6	IL17RB	Interleukin-17 receptor B
UniProtKB: Q8NAC3	IL17RC	Interleukin-17 receptor C
UniProtKB: P01583	IL1A	Interleukin-1 alpha
UniProtKB: P14778	IL1R1	Interleukin-1 receptor type 1
UniProtKB: P27930	IL1R2	Interleukin-1 receptor type 2
UniProtKB: Q01638	IL1RL1	Interleukin-1 receptor-like 1
UniProtKB: Q9HB29	IL1RL2	Interleukin-1 receptor-like 2
UniProtKB: P60568	IL2	Interleukin-2
UniProtKB: Q6UXL0	IL20RB	Interleukin-20 receptor subunit beta
UniProtKB: Q6UXL0	IL20RB	Interleukin-20 receptor subunit beta
UniProtKB: Q969J5	IL22RA2	Interleukin-22 receptor subunit alpha-2
UniProtKB: Q9H293	IL25	Interleukin-25
UniProtKB: P01589	IL2RA	Interleukin-2 receptor subunit alpha
UniProtKB: P01589	IL2RA	Interleukin-2 receptor subunit alpha
UniProtKB: Q8NI17	IL31RA	Interleukin-31 receptor subunit alpha
UniProtKB: O95760	IL33	Interleukin-33
UniProtKB: P05113	IL5	Interleukin-5
UniProtKB: Q01344	IL5RA	Interleukin-5 receptor subunit alpha
UniProtKB: P17301	ITGA2	Integrin alpha-2
UniProtKB: P05107	ITGB2	Integrin beta-2
UniProtKB: P18564	ITGB6	Integrin beta-6
UniProtKB: O60674	JAK2	Tyrosine-protein kinase JAK2
UniProtKB:Q9BX67	JAM3	Junctional adhesion molecule C
UniProtKB: P05412	JUN	Transcription factor AP-1
UniProtKB: O15054	KDM6B	Lysine-specific demethylase 6B
UniProtKB: P04264	KRT1	Keratin, type II cytoskeletal 1
UniProtKB: P18428	LBP	Lipopolysaccharide-binding protein
UniProtKB: P01130	LDLR	Low-density lipoprotein receptor
UniProtKB: P38571	LIPA	Lysosomal acid lipase/cholesteryl ester hydrolase
UniProtKB: Q5S007	LRRK2	Leucine-rich repeat serine/threonine-protein kinase 2
UniProtKB: P01374	LTA	Lymphotoxin-alpha
UniProtKB: P07948	LYN	Tyrosine-protein kinase Lyn
UniProtKB: P07948	LYN	Tyrosine-protein kinase Lyn
UniProtKB: P46734	MAP2K3	Dual specificity mitogen-activated protein kinase kinase 3
UniProtKB: P04201	MAS1	Proto-oncogene Mas
UniProtKB: Q8NEM0	MCPH1	Microcephalin
UniProtKB: P43490	NAMPT	Nicotinamide phosphoribosyltransferase
UniProtKB: Q16236	NFE2L2	Nuclear factor erythroid 2-related factor 2
UniProtKB: P19838	NFKB1	Nuclear factor NF-kappa-B p105 subunit
UniProtKB: Q9BYH8	NFKBIZ	NF-kappa-B inhibitor zeta
UniProtKB: P59044	NLRP6	NACHT, LRR and PYD domains-containing protein 6
UniProtKB: Q86UT6	NLRX1	NLR family member X1
UniProtKB: P46531	NOTCH1	Neurogenic locus notch homolog protein 1
UniProtKB: O15130	NPFF	Pro-FMRFamide-related neuropeptide FF
UniProtKB: P01160	NPPA	Natriuretic peptides A
UniProtKB: Q15761	NPY5R	Neuropeptide Y receptor type 5
UniProtKB: Q15761	NPY5R	Neuropeptide Y receptor type 5
UniProtKB: P21589	NT5E	5’-nucleotidase
UniProtKB: O60356	NUPR1	Nuclear protein 1
UniProtKB: O15527	OGG1	N-glycosylase/DNA lyase
UniProtKB: P35372	OPRM1	Mu-type opioid receptor
UniProtKB: P51575	P2RX1	P2X purinoceptor 1
UniProtKB: Q99572	P2RX7	P2X purinoceptor 7
UniProtKB: Q96KB5	PBK	Lymphokine-activated killer T-cell-originated protein kinase
UniProtKB: O75594	PGLYRP1	Peptidoglycan recognition protein 1
UniProtKB: Q96PD5	PGLYRP2	N-acetylmuramoyl-L-alanine amidase
UniProtKB: P48736	PIK3CG	Phosphatidylinositol 4,5-bisphosphate 3-kinase catalytic subunit gamma isoform
UniProtKB: Q9Y263	PLAA	Phospholipase A-2-activating protein
UniProtKB: P60201	PLP1	Myelin proteolipid protein
UniProtKB: P06746	POLB	DNA polymerase beta
UniProtKB: P37231	PPARG	Peroxisome proliferator-activated receptor gamma
UniProtKB: P42785	PRCP	Lysosomal Pro-X carboxypeptidase
UniProtKB: P28070	PSMB4	Proteasome subunit beta type-4
UniProtKB: P25105	PTAFR	Platelet-activating factor receptor
UniProtKB: O14684	PTGES	Prostaglandin E synthase
UniProtKB: O14684	PTGES	Prostaglandin E synthase
UniProtKB: P35354	PTGS2	Prostaglandin G/H synthase 2
UniProtKB: Q9ULZ3	PYCARD	Apoptosis-associated speck-like protein containing a CARD
UniProtKB: O95267	RASGRP1	RAS guanyl-releasing protein 1
UniProtKB: Q06330	RBPJ	Recombining binding protein suppressor of hairless
UniProtKB: Q9Y3P4	RHBDD3	Rhomboid domain-containing protein 3
UniProtKB: Q6R327	RICTOR	Rapamycin-insensitive companion of mTOR
UniProtKB: P05109	S100A8	Protein S100-A8
UniProtKB: P05109	S100A8	Protein S100-A8
UniProtKB: Q99500	S1PR3	Sphingosine 1-phosphate receptor 3
UniProtKB: Q15858	SCN9A	Sodium channel protein type 9 subunit alpha
UniProtKB: P18827	SDC1	Syndecan-1
UniProtKB: Q96EE3	SEH1L	Nucleoporin SEH1
UniProtKB: P16109	SELP	P-selectin
UniProtKB: P01008	SERPINC1	Antithrombin-III
UniProtKB: P36955	SERPINF1	Pigment epithelium-derived factor
UniProtKB: Q86VZ5	SGMS1	Phosphatidylcholine:ceramide cholinephosphotransferase 1
UniProtKB: P52569	SLC7A2	Cationic amino acid transporter 2
UniProtKB: P52569	SLC7A2	Cationic amino acid transporter 2
UniProtKB: Q15797	SMAD1	Mothers against decapentaplegic homolog 1
UniProtKB: P84022	SMAD3	Mothers against decapentaplegic homolog 3
UniProtKB: Q99835	SMO	Smoothened homolog
UniProtKB: O14543	SOCS3	Suppressor of cytokine signaling 3
UniProtKB: O75159	SOCS5	Suppressor of cytokine signaling 5
UniProtKB: Q9NYA1	SPHK1	Sphingosine kinase 1
UniProtKB: P10451	SPP1	Osteopontin
UniProtKB: P40763	STAT3	Signal transducer and activator of transcription 3
UniProtKB: P51692	STAT5B	Signal transducer and activator of transcription 5B
UniProtKB: Q9UEW8	STK39	STE20/SPS1-related proline-alanine-rich protein kinase
UniProtKB: Q9BXA5	SUCNR1	Succinate receptor 1
UniProtKB: P20366	TAC1	Protachykinin-1
UniProtKB: Q9NUY8	TBC1D23	TBC1 domain family member 23
UniProtKB: Q9UP52	TFR2	Transferrin receptor protein 2
UniProtKB: P21980	TGM2	Protein-glutamine gamma-glutamyltransferase 2
UniProtKB: P01033	TIMP1	Metalloproteinase inhibitor 1
UniProtKB: O60603	TLR2	Toll-like receptor 2
UniProtKB: O15455	TLR3	Toll-like receptor 3
UniProtKB: O00206	TLR4	Toll-like receptor 4
UniProtKB: Q9Y2C9	TLR6	Toll-like receptor 6
UniProtKB: Q9NYK1	TLR7	Toll-like receptor 7
UniProtKB: Q9NR97	TLR8	Toll-like receptor 8
UniProtKB: P01375	TNF	Tumor necrosis factor
UniProtKB: P21580	TNFAIP3	Tumor necrosis factor alpha-induced protein 3
UniProtKB: P20333	TNFRSF1B	Tumor necrosis factor receptor superfamily member 1B
UniProtKB: P20333	TNFRSF1B	Tumor necrosis factor receptor superfamily member 1B
UniProtKB: Q8NER1	TRPV1	Transient receptor potential cation channel subfamily V member 1
UniProtKB: Q8NER1	TRPV1	Transient receptor potential cation channel subfamily V member 1
UniProtKB: Q9HBA0	TRPV4	Transient receptor potential cation channel subfamily V member 4
UniProtKB: O60636	TSPAN2	Tetraspanin-2
UniProtKB: O75896	TUSC2	Tumor suppressor candidate 2
UniProtKB: P55089	UCN	Urocortin
UniProtKB: P22309	UGT1A1	UDP-glucuronosyltransferase 1-1
UniProtKB: Q70J99	UNC13D	Protein unc-13 homolog D
UniProtKB: P19320	VCAM1	Vascular cell adhesion protein 1
UniProtKB: P19320	VCAM1	Vascular cell adhesion protein 1
UniProtKB: Q9HC57	WFDC1	WAP four-disulfide core domain protein 1
UniProtKB: Q15942	ZYX	Zyxin

**Table 3 ijms-22-00388-t003:** CDEI testing results—EpiDermFT.

Data Set	Sample	Number of Profiles	*t*-Value	*p*-Value	CDEI
DMSO	Control (*C*)	3	0	1	-
UV	Untreated (*U*)	3	1.04	0.23	0.00
Extract #4	Treated (*T*)	5	0.67	0.50	0.22
Extract #12	Treated (*T*)	5	2.06	0.04	−0.33
Extract #8	Treated (*T*)	5	−0.19	0.85	0.69
Extract #13	Treated (*T*)	5	−1.04	0.30	0.00

**Table 4 ijms-22-00388-t004:** CDEI testing results—EpiOral.

Data Set	Sample	*t*-Value	*p*-Value	CDEI
DMSO	Control (C)	-	-	-
TNFα	Untreated (U)	−2.78	0.006	0.00
Extract #1	Treated (*T*)	0.86	0.39	0.53
Extract #7	Treated (*T*)	0.19	0.85	0.87
Extract #9	Treated (*T*)	−0.03	0.98	0.98
Extract #45	Treated (*T*)	−2.02	0.04	0.16
Extract #115	Treated (*T*)	−0.15	0.88	0.90
Extract #129	Treated (*T*)	−0.63	0.53	0.63
Extract #157	Treated (*T*)	−0.29	0.77	0.81
Extract #167	Treated (*T*)	−2.27	0.02	0.10
Extract #169	Treated (*T*)	−0.17	0.86	0.88

**Table 5 ijms-22-00388-t005:** CDEI testing results—EpiIntestinal.

Data Set	Sample	*t*-Value	*p*-Value	CDEI
DMSO	Control (C)	-	-	-
TNFα	Untreated (U)	2.43	0.016	0.00
Extract #1	Treated (*T*)	−1.37	0.17	0.28
Extract #7	Treated (*T*)	1.15	0.25	0.36
Extract #9	Treated (*T*)	1.43	0.15	0.26
Extract #45	Treated (*T*)	1.02	0.31	0.41
Extract #115	Treated (*T*)	0.21	0.84	0.84
Extract #129	Treated (*T*)	−0.56	0.58	0.63
Extract #130	Treated (*T*)	1.56	0.12	0.22
Extract #169	Treated (*T*)	1.75	0.08	0.16
Extract #274	Treated (*T*)	0.99	0.32	0.42

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
