# Peer review of "System, Method and Software for Calculation of a Cannabis Drug Efficiency Index for the Reduction of Inflammation"

_ijms, 2020, doi:10.3390/ijms22010388_

Round 1
Reviewer 1 Report
In this manuscript titled "System, Method and Software for Calculation of Cannabis Drug Efficiency Index in the Reduction of Inflammation", the authors develop a compound drug efficiency detection tool based on gene and signaling pathway (SPIA) data mining.
The approach and algorithms (supposed to be less time consuming) are in line with others reported in bioinformatics to be predictive and accurate ones.
The results (CDEI), despite their limitation to one natural product (cannabis extract) only, are sufficiently documented and acceptable.
More information about the graphical user interface (GUI) in supplementary information is expected.
The conclusion needs to be more extended.
For better clarity, the methodology has to be improved by a clear workflow.
Author Response
Reviewer: More information about the graphical user interface (GUI) in supplementary information is expected.
Our response
We have more information on the use of graphics in the software manual. We added the phrase “As detailed in the accompanying manual of the software…” in the manuscript.
Reviewer: The conclusion needs to be more extended.
Our response
We have extended the conclusion. We added the following: “Currently, CDEI algorithm was only used for the analysis of the efficacy of cannabis extracts in the reduction of inflammation. It remains to be shown whether it would also be able to rank cannabis extracts (or extracts of any other medicinal herb or synthetic drug) by the efficacy for other molecular processes, diseases or conditions. CDEI algorithm may also be an efficient tool to stratify patients in the clinical studies or clinical trials by their response to a drug, thus providing selection criteria upon transition from phase 2 to phase 3 or phase 4 clinical trials. It will also be very useful in pre-clinical experiments using various cell, tissue or animal models.”
Reviewer: For better clarity, the methodology has to be improved by a clear workflow.
Our response:
We have provided a new figure with the methodology
Reviewer 2 Report
The manuscript described the calculation of cannabis drug efficiency using the Cannabis drug efficiency index (CDEI) software. In many countries, the growth of cannabis is forbidden. The authors showed the method to predict the toxicities such as inflammation induction by cannabis drug using the CDEI software. Thus, this system will be useful for the treatment based on cannabis drug. Therefore, the manuscript is not too excellent to be published. In other words, the manuscript is so excellent that it should be published.
Comments
(1) I do not know whether the calculation are correct or not, because I am unfamiliar with the algorithm such as SPIA algorithm largely. How much were the CDEI results changed by chosen input data?
(2) How much precise were the CDEI results, compared to the observation?
(3) Will AI improve this system?
That is all.
Author Response
Comments
(1) I do not know whether the calculation are correct or not, because I am unfamiliar with the algorithm such as SPIA algorithm largely. How much were the CDEI results changed by chosen input data?
Our response
An important characteristic of the CDEI calculation is that CDEI is not affected by the properties of input data chosen. CDEI purely measures the efficacy of the cannabis drug by statistically comparing the cannabis-treated and untreated cases with the control.
In general, the question of model robustness for various approaches to gene-expression-based pathway activation assessment was studied in detail using perturbation and parameter stiffness theory in the past decade. Acceptable robustness was obtained in PMID24723936, PMID28849557, PMID28825872.
(2) How much precise were the CDEI results, compared to the observation?
Our response
We are not sure what is asked by the reviewer. CDEI calculations were done on multiple (over 20) extracts on three different (different tissues, treatment etc.) experiments and in all cases the software was able to clear (and significantly) rank the extracts in their efficacy.
(3) Will AI improve this system?
Our response:
We added the following sentence: “Pathway analysis has the potential to improve performance by applying machine learning methods. For example, the representative genes that represent a pathway can be selected by feature selection methods and pathways can be further ranked or weighted based on the representative genes using classification methods (Zhang W, Emrich S, Zeng E. A two-stage machine learning approach for pathway analysis. IEEE International Conference on Bioinformatics and Biomedicine 2010; 274-279).”
In general, however, the problem with AI/ML is that any approach to gene-expression-based pathway activation assessment implies much more parameters, which may be optimized, in comparison with the number of data examples, which may be used for training the AI/ML models. This means that such optimization using AI/ML would be almost inevitably overtrained, and alternative approaches, e.g. gradient descent etc. seem more preferable for this improvement. However, this improvement is a principally different research topic, and most approaches to gene-expression-based pathway activation assessment (including all approaches cited in the current paper), still have not elaborated an effective and robust parameter optimization scheme.
This manuscript is a resubmission of an earlier submission. The following is a list of the peer review reports and author responses from that submission.
Round 1
Reviewer 1 Report
The authors describe the development of a method and software to analyze various cannabis extracts' efficiency to detect antiinflammatory properties. The method uses the transcriptomics data to identify the Cannabis extract that is able to induce the expression of an antiinflammatory genotype. Based on differences in various extracts' efficiency in restoring transcriptome changes to the pre-inflammation state, a Cannabis Drug Efficiency Index (CDEI) has been determined.
The work presents a new and interesting analytical approach providing an important contribution to personalized medicine.
Comments:
- The authors state that the method can be extended to all other drugs; therefore they should clarify the motivations to chose the Cannabis antiinflammatory action to develop their method and software.
- It would be appropriate to report the genes considered for the analysis and the ratio of expression for each sample, also expressed as Fold change.
- It is unclear whether the extract's efficiency is only related to the CDEI index or the p-value and whether the latter is related to the significance of the data. The authors should explain the following: "In the first experiment, we induced inflammation in human EpiDermFT 3D skin tissues by exposing it to UVC. Analysis of the samples from the first experiment revealed that extract #8 is the most efficient extract in restoring the transcriptome response after the UVC exposure (Table 1). Extract #4 was less efficient, whereas extract #13 was not efficient. Extract #12 was actually harmful, as it increased the UVC-induced changes in the transcriptome”
- In Table 1, extract 8 is characterized by the best CDEI value indicating the highest antiinflammatory efficiency; however, the p-value (not lower than 0.05) seems to be not significant…. The authors should clarify the correlation between the two parameters.
- It would be interesting to confirm the results using statistical tests and a trans-omic approach supporting the selected pathways' involvement.
Author Response
Reviewer 1
The work presents a new and interesting analytical approach providing an important contribution to personalized medicine.
Our response:
Thank you for the kind words.
Reviewer 1
- The authors state that the method can be extended to all other drugs; therefore they should clarify the motivations to chose the Cannabis antiinflammatory action to develop their method and software.
Our response:
We have added the rational. The main reason for the development of the software and use for the evaluation of the anti-inflammatory properties of cannabis was because we were preparing clinical trial and wanted to develop the stratification mechanism for selection of patients – responders versus non-responders.
We have added the following to the Intro section “Such method can potentially be used for the detection of the efficiency of various other botanical extracts or single compounds, although the versatility of the method will have to be confirmed using other data sets.”
Reviewer 1
- It would be appropriate to report the genes considered for the analysis and the ratio of expression for each sample, also expressed as Fold change.
Our response:
We agree with the reviewer and added the data on the genes used for the analysis of expression. This is currently Table 1, Figure 2 and Supplementary File 1.
Reviewer 1
- It is unclear whether the extract's efficiency is only related to the CDEI index or the p-valueand whether the latter is related to the significance of the data. The authors should explain the following: "In the first experiment, we induced inflammation in human EpiDermFT 3D skin tissues by exposing it to UVC. Analysis of the samples from the first experiment revealed that extract #8 is the most efficient extract in restoring the transcriptome response after the UVC exposure (Table 1). Extract #4 was less efficient, whereas extract #13 was not efficient. Extract #12 was actually harmful, as it increased the UVC-induced changes in the transcriptome”
Our response
p-value here is not related to the significance of the data. Induction of inflammation is reflected by the changes in the expression of various pro-inflammatory genes. The CDEI reflects the reversal of this change. The closer to 1.0, the more complete is reversion, with 1.0 would be complete restoration of transcriptome changes. Negative CDEI (as in the case of the extract #12) indicates that changes in transcriptome have worsened – they further changed in the direction of pro-inflammation.
- In Table 1,extract 8 is characterized by the best CDEI value indicating the highest antiinflammatory efficiency; however, the p-value (not lower than 0.05) seems to be not significant…. The authors should clarify the correlation between the two parameters.
Our response
p-value here is not related to the significance of the data. We have clarified this now.
- It would be interesting to confirm the results using statistical tests and a trans-omic approach supporting the selected pathways' involvement.
Our response
In the future, we will perform proteomics data and attempt to validate these experimental sets.
Reviewer 2 Report
The submitted manuscript provides a new methods aimed at identifying the efficiency of various cannabis extracts in term of anti inflammatory properties in human tissues. The manuscript is interesting, but appears quite pivotal. The authors have to test their methods on a larger set of samples and there is the need to upstream know the different (in any) and exact composition of their cannabis extract to validate their methods, thus providing a new interesting methodology for cannabis extract selection.
Furthermore, I do not find the manuscript suitable to IJMS, being the manuscript methodological.
minor points:
- the title is too much general, anti inflammatory properties only have been investigated
- in the presentation of results the differences between high and low cannabis extract are not clear
- abbreviations has not been consistently defined and used all over the main text
- Source of tissue/cell lines in missing
Author Response
The submitted manuscript provides a new methods aimed at identifying the efficiency of various cannabis extracts in term of anti inflammatory properties in human tissues. The manuscript is interesting, but appears quite pivotal. The authors have to test their methods on a larger set of samples and there is the need to upstream know the different (in any) and exact composition of their cannabis extract to validate their methods, thus providing a new interesting methodology for cannabis extract selection.
Our response
We thank you the reviewer for their opinion. We agree that our manuscript is quite pivotal (essential and critically important).
Although it would of cause be interesting to test our methods on a larger set of data, the conclusions that were drown here were based on three independent data sets representing different tissues and different extracts.
As to the composition of the extracts, we are not sure why is it relevant. The task in hand was to identify which extract is the best to use for specific condition. Potential correlation between composition and the activity of the extract was not the goal of this work. Nevertheless, we have attempted to do such correlation for the data we had in hand; we did not find any correlation between cannabinoids composition and the activity of the extracts. This can be due to the modulatory effects of many other molecules in the botanical extract. This is why CDEI is so important for the analysis of the efficiency of the extracts.
Furthermore, I do not find the manuscript suitable to IJMS, being the manuscript methodological.
Our response
We are not sure why it would not be suitable to IJMS. While we are presenting the methodology (software development), we present molecular data for three experiments.
Reviewer 2.
minor points:
- the title is too much general, anti inflammatory properties only have been investigated
Our response
We have changed the title. It now reads: “SYSTEM, METHOD AND SOFTWARE FOR CALCULATION OF CANNABIS DRUG EFFICIENCY INDEX IN THE REDUCTION OF INFLAMMATION”
- in the presentation of results the differences between high and low cannabis extract are not clear
Our response
Not sure what is asked here. What does high and low cannabis extract mean? Do you mean high and low activity? We have added additional information as to the source of extracts. As mentioned above, correlation analysis did not show any correlation between the level of cannabinoids and the activity.
- abbreviations has not been consistently defined and used all over the main text
Our response
We have taken care of this.
- Source of tissue/cell lines in missing
Our response
We have added this information.
Round 2
Reviewer 2 Report
I am really sorry, but I confirm my previous report.
The manuscript is interesting, but quite pivotal and I do not find it suitable to IJMS. Furthermore, since the extracts of several cannabis cultivars have different activity, in my opinion it is mandatory to know what is the exact composition of the tested extracts.